# Effects of Cesarean Section and Vaginal Delivery on Abdominal Muscles and Fasciae

**DOI:** 10.3390/medicina56060260

**Published:** 2020-05-27

**Authors:** Chenglei Fan, Diego Guidolin, Serena Ragazzo, Caterina Fede, Carmelo Pirri, Nathaly Gaudreault, Andrea Porzionato, Veronica Macchi, Raffaele De Caro, Carla Stecco

**Affiliations:** 1Department of Neurosciences, Institute of Human Anatomy, University of Padua, 35121 Padua, Italy; yutianfan1218@163.com (C.F.); diego.guidolin@unipd.it (D.G.); serena.raga@gmail.com (S.R.); caterina.fede@unipd.it (C.F.); carmelop87@hotmail.it (C.P.); andrea.porzionato@unipd.it (A.P.); veronica.macchi@unipd.it (V.M.); rdecaro@unipd.it (R.D.C.); 2Faculty of Medicine and Health Sciences, School of Rehabilitation, University of Sherbrooke, 3001, 12e Avenue Nord, Sherbrooke, QC J1H 5N4, Canada; nathaly.gaudreault@usherbrooke.ca

**Keywords:** caesarean section, vaginal delivery, fascia, abdominal muscles, pain, ultrasound

## Abstract

*Background and objectives*: Possible disorders after delivery may interfere with the quality of life. The aim of this study was to ascertain whether abdominal muscles and fasciae differ in women depending on whether they experienced transverse cesarean section (CS) or vaginal delivery (VA) in comparison with healthy nulliparous (NU). *Materials and methods*: The thicknesses of abdominal muscles and fasciae were evaluated by ultrasound in 13 CS, 10 VA, and 13 NU women (we examined rectus abdominis (RA); external oblique (EO); internal oblique (IO); transversus abdominis (TrA); total abdominal muscles (TAM = EO + IO + TrA); inter-rectus distance (IRD); thickness of linea alba (TLA); rectus sheath (RS), which includes anterior fascia of RS and posterior fascia of RS (P-RS); loose connective tissue between sublayers of P-RS (LCT); abdominal perimuscular fasciae (APF), which includes anterior fascia of EO, fasciae between EO, IO, and TrA, and posterior fascia of TrA). Data on pain intensity, duration, and location were collected. *Results*: Compared with NU women, CS women had wider IRD (*p* = 0.004), thinner left RA (*p* = 0.020), thicker right RS (*p* = 0.035) and APF (left: *p* = 0.001; right: *p* = 0.001), and IO dissymmetry (*p* = 0.009). VA women had thinner RA (left: *p* = 0.008, right: *p* = 0.043) and left TAM (*p* = 0.024), mainly due to left IO (*p* = 0.027) and RA dissymmetry (*p* = 0.035). However, CS women had thicker LCT (left: *p* = 0.036, right: *p* < 0.001), APF (left: *p* = 0.014; right: *p* = 0.007), and right IO (*p* = 0.028) than VA women. There were significant correlations between pain duration and the affected fasciae/muscles in CS women. *Conclusions*: CS women showed significant alterations in both abdominal fasciae and muscle thicknesses, whereas VA women showed alterations mainly in muscles. Thinner RA and/or dissymmetric IO, wider IRD, and thicker LCT and APF after CS may cause muscle deficits and alteration of fascial gliding, which may induce scar, abdominal, low back, and/or pelvic pain.

## 1. Introduction

Cesarean section (CS) is nowadays one of the most common surgical techniques performed on women. According to data from 150 countries, currently 18.6% (6%–27.2%) of all births occur by CS. Based on data from 121 countries, the global average CS rate increased 12.4% (6.7%–19.1%), with an average annual rate of increase of 4.4%, between 1990 and 2014 [1]. Increased maternal age, better social and economic conditions, and the mistaken belief that vaginal delivery (VA) may be dangerous make CS a very common surgical procedure [2]. However, possible disorders after CS may interfere with the quality of life. Many authors have focused on maternal complications following CS, and the debate about its benefits and complications is ongoing. CS scar syndrome refers to a set of symptoms, including pelvic pain (PVP) and dysmenorrhea caused by an abnormally healed CS scar [3]. Many studies have shown a positive association between CS and some complications such as scar pain (SP) [4], PVP [5], abdominal pain (ABP) [6], and low back pain (LBP) [7], determined by questionnaires, interviews, or pelvic floor examination.

To our knowledge, previous research has generally studied the changes in some abdominal muscles following CS and/or VA [8], whereas no prior studies have addressed the topic from the anatomical and fascial points of view, considering especially the fasciae, in order to understand how these tissues respond to different delivery modes and how alterations may be related to SP, ABP, LBP, and PVP. Fascia has a multitude of functions, including transmission of force, coordination of movements, stability, and proprioceptive communications throughout the body, promoting sliding and reducing the friction associated with motion [9,10]. Any trauma, surgery, or overuse syndrome of structures far from the injured region may alter the sliding system within the fascial plane [11]. Therefore, CS scars may lead to symptoms in the abdominal region or in a distal area due to fascial continuity.

The first aim of this study was to study whether the thickness of abdominal fasciae and muscles differs between subjects who experiences different delivery models (CS, VA) and healthy nulliparous individuals (NU). Secondly, we studied possible correlations between changes of the abdominal fasciae or muscles and pain.

## 2. Materials and Methods

### 2.1. Participants

We recruited 13 NU women (mean age 27.08 ± 14.23 years; mean body mass index (BMI): 21.64 ± 1.44 kg/cm^2^), 13 primiparous women who had transverse CS (mean age 41.69 ± 6.11 years; BMI: 23.70 ± 2.50 kg/cm^2^), 10 VA women (mean age 47.00 ± 15.19; years BMI: 21.52 ± 2.77 kg/cm^2^) according to the following criteria: first-born child, delivery by transverse CS or VA between 2001 and 2019, at least two years after delivery, not having taken part in any rehabilitation program, and not having abdominal surgery or trauma in the post-partum period. We chose only women with one CS or one VA. All subjects gave their informed consent for inclusion before they participated in the study. The study was conducted in accordance with the Declaration of Helsinki, and the protocol was approved by the Ethics Committee of University of Padua.

### 2.2. Procedure

After giving their informed and written consent, all participants had their weight and height measured and underwent ultrasound imaging evaluation of the abdominal muscles and related fasciae, while lying supine in a resting position. Primiparous participants (CS, VA) were then asked to complete a questionnaire on one-time-point data about pain intensity, duration, and location felt to during the post-partum period. Correlations among thickness of abdominal muscle and fasciae, pain intensity, and duration were examined.

### 2.3. Ultrasound Measurement of Abdominal Muscles and Fascial Morphometry

Ultrasound measurements were obtained from participants at the end of the exhalation phase in three probe locations (linea alba, LA; rectus abdominis muscle region, RA; axillary line, AL) on both sides of the abdominal wall. Abdominal muscles and fascial thickness were measured on Esaote MyLab Seven ultrasound machine (Esaote SpA, Genova, Italy) with 37 mm linear-array transductors, 6–18 MHz, by the same experienced operator in physical and rehabilitation medicine with 5 years’ experience in ultrasound skeletal muscle imaging (Figure 1), since our previous study confirmed the good inter-rater reliability indicating that ultrasound imaging is a reliable instrument for evaluating abdominal muscles/fasciae thickness [12]. The parameters for all three probe locations included:For LA, 2 cm above the umbilicus: thickness of linea alba (TLA) and inter-rectus distance (IRD) (Figure 1A).For RA, at the same level of the umbilicus: thickness of RA muscle, anterior fascia of rectus sheath (A-RS), posterior fascia of rectus sheath (P-RS), loose connective tissue between sublayers of the P-RS (LCT) (Figure 1B).For AL, at the same level of the umbilicus: thickness of external oblique muscle (EO), internal oblique muscle (IO), transverse abdominis muscle (TrA), total abdominal muscles (TAM = EO + IO + TrA); anterior fascia of EO (FEO), fasciae between EO and IO (FEO/IO), fasciae between OI and TrA (FOI/TrA), posterior fascia of transverse abdominis muscle (FTrA), and abdominal perimuscular fasciae (APF = FEO + FEO/IO + FIO/ TrA+ FTrA) (Figure 1C).

### 2.4. Pain Questionnaire

After ultrasound imaging evaluation, the participants were asked to complete a short questionnaire to collect data about pain intensity, duration, and location in the post-partum period. Pain intensity was measured with a visual analog scale (VAS) (mild: ≤3.4, moderate: 3.5 to 7.4, severe: ≥7.5) [13]. They were asked how bad the pain was in four locations: on the scar and in the areas of the abdominal wall, lower back, and pelvic floor. They were also asked how long this pain had lasted after CS and VA (acute pain: <1month, subacute pain: 1–3 months, chronic pain: >3 months) [14].

### 2.5. Statistical Analysis

All data management and statistical analyses were performed with IBM SPSS version 25 (IBM Corp. Released 2017. IBM SPSS Statistics for Windows, Version 25.0. Armonk, NY, USA). The first purpose was to determine whether variables were or were not normally distributed by the Shapiro–Wilks test. Since all groups were found to have normal distribution, univariate correlation analyses, based on Pearson’s correlation coefficients(r) (normally distributed data) were used to examine the relationship between ultrasound parameters and age and BMI in all subjects. Analyses of covariance (normally distributed data) were used to compare the thickness of the abdominal muscles and fasciae. Secondly, variables related to the right and left sides were compared by a paired Student’s t-test, since the distribution of the variables appeared to be normal on each side. Thirdly, for CS and VA subjects, relationships among abdominal muscles, fasciae morphology, and pain intensity and duration (SP, ABP, LBP, PVP) were examined with Spearman’s correlation coefficients (rs). The level of significance was set at *p* ≤ 0.05.

## 3. Results

The baseline characteristics of the subjects included in this study are presented in Table 1. CS and VA women had similar age, slightly lower for VA women; NU women were younger than CS (*p* = 0.002) and VA women (*p* = 0.001), because it was difficult to recruit matched-age NU women according to the enrolled criteria. In addition, right (R)-RA, left (L)-RA, and IRD were correlated with age (r = −0.54, −0.56, and 0.55, *p* < 0.001). R-RS, L-RS, R-APF, L-APF, TLA were correlated with BMI (r = 0.38, 0.38, 0.39, 0.34, −0.33, *p* < 0.05). Accordingly, analyses of covariance with age as a covariate for R-RA, L-RA, and IRD and BMI as a covariate for R-RS, L-RS, R-APF, L-APF, and TLA were performed to compare groups.

### 3.1. Muscular Morphometry

Descriptive data for RA, total abdominal muscle (EO + IO + TrA), and individual muscle thickness (EO, IO, TrA) for the three groups are listed in Table 2. In the rectus abdominal muscle region, compared with NU subjects, RA was significantly thinner on the left side in CS women (*p* = 0.020) and on both sides in VA women (left: *p* = 0.008, right: *p* = 0.043). RA in VA subjects presented a significant difference between the left and the right sides (*p* = 0.035). In the axillary line, compared with NU women, L-TAM was thinner (*p* = 0.024) in VA women, mainly due to L-IO (*p* = 0.027). R-IO in CS subjects was significantly thicker (*p* = 0.028) compared with VA subjects. IO in CS women showed a significant difference between left and right sides (*p* = 0.009).

### 3.2. Fascial Morphometry

Descriptive data for IRD, TLA, RS, LCT, and APF for the three groups are listed in Table 3. At the linea alba, compared with NU women, CS women had a wider IRD (CS: 25.55 ± 6.48 mm, NU: 12.69 ± 6.28 mm, VA: 20.75 ± 7.41 mm; *p* = 0.004). The analysis of TLA revealed no differences in the three groups. In the rectus abdominal muscle region, compared with NU women, R-RS was significantly thicker in CS women (*p* = 0.035), whereas L-LCT and R-LCT were significantly thicker in CS than in VA subjects (left: *p* = 0.036, right: *p* <0.001). In the axillary line, L-APF and R-APF in CS subjects were significantly thicker compared with NU (left: *p* = 0.001; right: *p* = 0.001) and VA women (left: *p* = 0.014; right: *p* = 0.007).

### 3.3. Pain Questionnaire

A series of questions concerning physical post-partum health were submitted to the recruited women, asking if they had experienced complications such as SP, ABP, LBP, or PVP. We obtained the following results for CS women: all subjects (100%) complained of SP, defined as annoyance and pain at the wound site (15% mild, 54% moderate, 31% severe), that lasted less than 1 month for 5/13 of them, from 1 to 3 months for 4/13, and for more than 6 months for 4/13. In addition, 12/13 subjects (92%) suffered from ABP, characterized by more widespread pain in the abdominal wall even in sites distant from the wound, which was mild for 31% of the subjects, moderate for 46%, and severe for 15% of them. We also found that 69% of the subjects experienced pain for less than one month from delivery, 15% for 1–3 months, and only one subject for over 6 months (8%). Moreover, 10/13 subjects (77%) complained of LBP or lumbar rachis pain, which was mild for 31% of them, moderate for 38%, and severe for 8%. For 5/13 subjects (39%), pain lasted for less than 1 month, for 3/13 from 1 to 3 months, and for 2/13 longer than 6 months. As regards PVP, defined as feeling pain in the lower abdomen and the perineal or genital area, 8/13 subjects (61%) suffered from it, 5 in mild form, and 3 in moderate form. Finally, 6/13 of the subjects experienced pain for less than 1 month, 1/13 subjects from 1 to 3 months, and 1/13 subjects for more than 6 months (Figure 2). Concerning VA women, only one subject complained of PVP in moderate form for 1 week; the others had no pain.

### 3.4. Correlation Between the Thickness of the Affected Abdominal Muscles and Fasciae and Clinical Measures

Spearman’s coefficients of correlation between the thickness change of abdominal muscles and fasciae and pain duration were determined. The duration of SP correlated moderately with IRD (rs = 0.69, *p* = 0.008). The duration of ABP correlated moderately with R-RS, L-RS, and L- LCT (rs = 0.56, *p* = 0.046, rs = 0.61, *p* = 0.027, rs = 0.68, *p* = 0.009). the duration of LBP correlated moderately with R-RS and L-APF (rs = 0.59, *p* = 0.034, rs = 0.58, *p* = 0.038). The duration of PVP correlated moderately with L- RA (rs = −0.55, *p* < 0.05). However, there were no significant correlations between the thickness of the affected abdominal muscles and fasciae and pain intensity.

## 4. Discussion

To our knowledge, this is the first study considering the abdominal muscles and fasciae in individuals who experienced CS and VA. The results indicate that CS and VA effects on abdominal muscles and fasciae are not homogeneous. CS women showed statistically significant alterations in both abdominal fasciae (wider IRD; thicker R-RS, L-APF, R-APF) and muscles (thinner L-RA; dissymmetry of IO), whereas VA women showed alterations mainly in muscles (thinner RA and L-TAM, mainly due to L-IO; dissymmetry of RA). Although the fasciae in VA subjects were slightly thicker than in NU subjects, the results were not statistically significant.

Our results are consistent with the findings of Weis et al. [8] which demonstrated that VA subjects had thinner RA and IO than nulliparous women, whereas they presented no significant differences in EO and TrA. Whittaker et al. [15] also found that patients with lumbopelvic pain had thinner total abdominal muscle, RA, and thicker perimuscular connective tissue. In addition, correlation coefficients between L-RA and PVP duration were found to be significant. Based on our findings and those of Weis et al., it would appear that RA and IO are seriously affected by pregnancy and delivery. As the abdomen becomes protuberant during pregnancy, the abdominal muscles, fasciae, and subcutaneous tissue are subjected to internal pressure due to increased abdominal volume [16]. The abdominal tissues become thinner until delivery, due to the application of a mechanical force. The lower portion of the IO fibers, like the RA, may be affected, and as such, a change in muscle thickness may account for a change in overall strength, affecting muscle function and perhaps contributing to lumbopelvic pain [8]. Another possible cause of alterations in abdominal wall is the presence of high levels of β-estradiol in the abdominal fasciae during pregnancy. Our previous in vitro study demonstrated that fascial cells can modulate the production of some components of the extracellular matrix according to hormone levels [16]. Therefore, hormone levels, together with biomechanical and delivery factors, could modify the morphology of abdominal muscles and fasciae [17,18]. We found a statistically significant dissymmetry of RA in VA women and of IO in CS women, which leads us to believe that the cause of this muscular dissymmetry is pregnancy itself and/or delivery. The important roles played by the abdominal muscles in postural control, spinal stabilizationm and movement coordination are well known. The contribution of abdominal insufficiency, imbalance, and/or weakness of the trunk musculature to LBP or PVP in pregnant women has been reported [19]. However, in our study, both CS and VA women had thinner RA and muscle dissymmetry, while most of the CS subjects suffered from SP, ABP, LBP, and PVP, which suggests that RA and IO dissymmetry may contribute to pain in CS women but they are not the only factors involved.

Although the fascia has been hypothesized to play a role in the pathogenesis of chronic pain, no investigation quantitatively evaluating the fasciae in CS or VA subjects can be found in the literature. From the fascial point of view, abdominal muscles are in continuity with the thoracolumbar fascia and the pelvic floor [20,21,22,23]. It has been demonstrated that they work with great synergy [24], guaranteed by fascial continuity. Therefore, all the abdominal muscles and fasciae work synergistically and simultaneously. In our study, CS women suffering from pain showed more alterations in the fasciae than VA women, which leads us to believe that the cause of these different effects is probably the different delivery mode, since the CS protocol produces a scar and mainly influences the fasciae rather than the abdominal muscles. IRD-associated abdominal weakness may be related to a weak connective tissue, which may then affect thoracic, abdominal, and pelvic dynamics, resulting in the application of a force from the diaphragm in the thorax and the abdomen to the pelvic floor muscles, according to the theory of pelvic dynamics. Thus, IRD may be one of the risk factors for female pelvic floor dysfunction, characterized by painful urination, constipation or bowel strains, LBP, and PVP [25]. According to Benjamin et al., a widening >2.7 cm at the level of the umbilicus is considered a pathological diastasis of RA [26]. Rett et al. [27], instead, consider IRD to be significant when it is more than 2 cm. However, in our study, CS women presented a wider IRD (CS: 25.55 ± 6.48 mm, NU: 12.69 ± 6.28 mm, VA: 20.75 ± 7.41 mm) and duration of SP, which correlated moderately with IRD. It has been found that IRD incidence was significantly higher in CS than in VA women [25]. CS can cause a greater risk for increased IRD; in addition, a wider IRD may contribute to altering the normal gliding between fascial layers, weakening the abdominal wall.

Our previous review recommended that measuring LCT between dense fibrous sublayers could be helpful to widen our understanding of iliotibial band syndrome and improve care for the patients [28]. In this study, our data showed that LCT and APF in CS women were significantly thicker than in VA women. In addition, there were moderate significant correlations between LCT and ABP duration and between APF and LBP duration. Scar formation after CS causes soft tissue adhesion, which may change the thickness of the abdominal muscles and subcutaneous tissue [16] as well as of the deep fasciae and disrupt normal fascia architecture. When the dermis and fasciae are affected by a scar, the sliding structure of the fasciae is altered, and when the scar tissues are not capable of adapting to the new stressor, their function is impaired. A fascial scar may lead to a dysfunction of the fascia itself, and symptoms may arise both directly in the scar area and at a distance from it, due to fascial continuity [29]. Our clinical data confirmed that CS women reported not only SP, but also pain in distal regions such as ABP, LBP, and PVP.

Ultrasound, therefore, is a good instrument in providing immediate feedback about the status of the abdominal muscles and fasciae. It would be useful to monitor women before pregnancy to assess their abdominal wall status and then to encourage them to attend appropriate exercise programs in the post-partum period, once allowed by the pelvic floor conditions, to restore good muscle functioning and to prevent the onset of complications in the following years.

This study has several limitations. First, being a retrospective study, many factors could not be controlled. We do not know about women’s health conditions and abdominal status before pregnancy; we are not sure that CS scar or VA is the main cause of the alterations in the thickness of abdominal fasciae and/or muscles. Future research needs to examine the thickness of abdominal fasciae and/or muscles before and after CS or VA. Second, the current data did not include other potential mediators such as psychological and/or pathoanatomical factors. Third, the number of enrolled patients and controls was small; therefore, it is difficult to generalize these study results to all women who experienced CS and VA. Further studies need to be conducted in larger and more diverse subject populations and include ultrasound analysis of fascial and/or muscular features of the lumbar and pelvic region.

## 5. Conclusions

Contrary to public opinion, according to which CS preserves women from post-partum complications, our study shows that CS, especially scar formation, may be one of the most important co-factors in developing muscle deficit and asymmetries and altering sliding within the fascial plane, which could, in turn, be the direct or indirect cause of SP, ABP, LBP, and PVP. Our study confirms the concept of a ’theory of a whole-body fascial linkage’ from the clinical point of view [20], which may help us to better understand the clinical symptoms of musculoskeletal pain, such as LBP resulting from the abdominal (e.g, CS, abdominal surgery) or pelvic region due to fascial continuity, and to develop appropriate treatments. Conversely, pelvic or abdominal symptoms may be present in low back disorders.

## Figures and Tables

**Figure 1 medicina-56-00260-f001:**
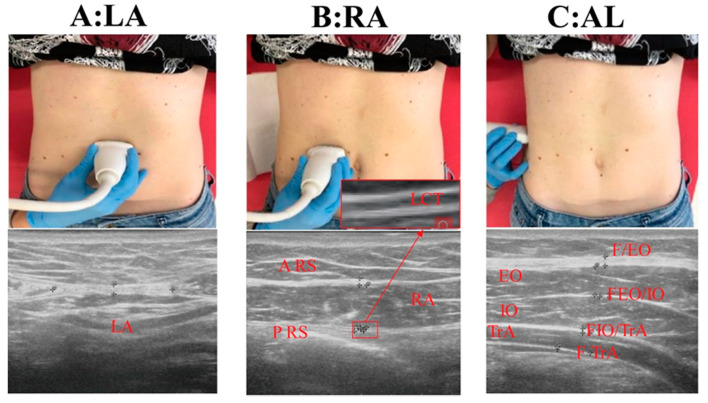
(**A**): LA: linea alba; (**B**): RA: rectus abdominal muscle; (**C**): AL: axillary line; A-RS: anterior fascia of rectus sheath; P-RS: posterior fascia of rectus sheath; LCT: loose connective tissue between sublayers of P-RS: EO: external oblique muscle; FEO: anterior fascia of external oblique muscle; IO: internal oblique muscle; TrA: transversus abdominis muscle; FTrA: posterior fascia of transversus abdominis muscle; FEO/IO: fasciae between external oblique and internal oblique muscle; FIO/TRA: fasciae between internal oblique and transversus abdominal muscle; TAM: total abdominal muscles = EO + IO + TrA; APF: abdominal perimuscular fasciae = FEO + FEO/IO + FIO/TrA + FTrA.

**Figure 2 medicina-56-00260-f002:**
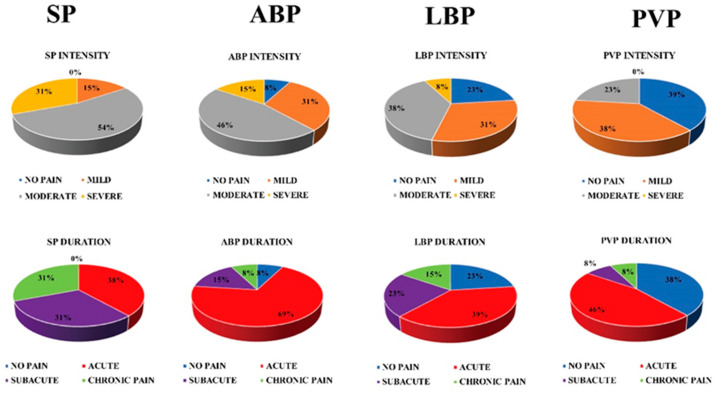
Results of the CS pain questionnaire. SP: scar pain; ABP: abdominal pain; LBP: low back pain; PVP: pelvic pain.

**Table 1 medicina-56-00260-t001:** Descriptive characteristics of the study. Values are presented as mean ± SD.

Characteristic	Group	N	Mean ± SD	CS vs. NU (*p*)	CS vs. VA (*p*)	VA vs. NU (*p*)
Age, y	CS	13	41.69 ± 6.11	0.002 *	0.738	0.001 *
	VA	10	47.00 ± 15.19			
	NU	13	27.08 ± 14.23			
BMI, kg/m^2^	CS	13	23.70 ± 2.50	0.022 *	0.030 *	0.446
	VA	10	21.52 ± 2.77			
	NU	13	21.64 ± 1.44			

Abbreviations: BMI, body mass index; CS, cesarean section group; VA, vaginal delivery group; NU, nulliparae group; N, number. * Difference between groups is statistically significant (*p* < 0.05).

**Table 2 medicina-56-00260-t002:** Muscle ultrasound parameter measurements. Values are presented as mean ± SD mm.

Position	Muscle	CS	NU	VA	CS vs. NU (*p*)	CS vs. VA (*p*)	NU vs. VA (*p*)
RA	L-RA	8.05 ± 1.75	10.79 ± 1.91	7.25 ± 1.42 ^§^	0.020 *	1.000	0.008 *
	R-RA	8.50 ± 1.91	10.95 ± 1.87	7.89 ±1.33 ^§^	0.061	1.000	0.043 *
AL	L-EO	5.52 ± 0.89	6.28 ± 1.13	5.51 ± 1.02	0.203	1.000	0.246
	R-EO	5.51 ± 1.28	6.88 ± 1.91	5.33 ± 1.49	0.106	1.000	0.082
	L-IO	6.29 ± 1.74 ^†^	6.76 ± 2.1	4.80 ± 0.63	1.000	0.126	0.027 *
	R-IO	7.62 ± 2.62 ^†^	7.13 ± 1.46	5.38 ± 1.29	1.000	0.028 *	0.116
	L-TRA	3.45 ± 0.64	3.49 ± 0.62	3.21 ± 1.20	1.000	1.000	1.000
	R-TRA	3.70 ± 1.16	3.48 ± 0.61	3.24 ± 1.01	1.000	0.775	1.000
	L-TAM	15.27 ± 2.63	16.53 ± 2.79	13.52 ± 2.00	0.640	0.331	0.024 *
	R-TAM	16.83 ± 4.42	17.49 ± 3.18	13.95 ± 2.68	1.000	0.191	0.073
		*p*^†^ = 0.009		*p*^§^ = 0.035			

Abbreviations: CS, cesarean section group, VA, vaginal delivery group, NU, nulliparous group; RA, rectus abdominal muscle; AL, axillary line; L-, left; R-, right; EO, external oblique; IO, internal oblique; TrA, transversus abdominis; TAM, total abdominal muscle. * Difference between groups is statistically significant (*p* < 0.05); †, § difference between sides is statistically significant (*p* < 0.05).

**Table 3 medicina-56-00260-t003:** Fascial ultrasound parameter measurements. Values are presented as mean ± SD mm.

Position	Fascia	CS	NU	VA	CS vs. NU(*p*)	CS vs. VA(*p*)	Nu vs. VA(*p*)
LA	IRD	25.55 ± 6.48	12.69 ± 6.28	20.75 ± 7.41	0.004 *	0.104	0.679
	TLA	1.82 ± 0.44	2.34 ± 0.74	2.42 ± 0.93	0.635	0.495	1.000
RA	L-RS	2.64 ± 0.50	2.04 ± 0.57	2.26 ± 0.48	0.102	0.841	0.912
	R-RS	2.37 ± 0.31	1.97 ± 0.31	2.24 ± 0.27	0.035 *	1.000	0.087
	L-LCT	0.46 ± 0.12	0.40 ± 0.10	0.34 ± 0.10	0.878	0.036 *	0.319
	R-LCT	0.42 ± 0.07	0.37 ± 0.05	0.30 ± 0.05	0.068	<0.001 *	0.050
AL	L-APF	3.44 ± 0.58	2.55 ± 0.44	2.75 ± 0.36	0.001 *	0.014 *	0.996
	R-APF	3.25 ± 0.40	2.63 ± 0.32	2.70 ± 0.31	0.001 *	0.007 *	1.000

Abbreviations: CS, cesarean section group, VA, vaginal delivery group, NU, nulliparous group; LA, linea alba; TLA, thickness of linea alba; RA, rectus abdominal muscle; L-, left; R-, right; RS, rectus sheath; LCT, loose connective tissue between sublayers of posterior fascia of RS; AL, axillary line; APF, abdominal perimuscular fasciae. * Difference between groups is statistically significant (*p* < 0.05).

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
