# Peer review of "Effects of Cesarean Section and Vaginal Delivery on Abdominal Muscles and Fasciae"

_medicina, 2020, doi:10.3390/medicina56060260_

Round 1
Reviewer 1 Report
The aim of this study was to ascertain whether abdominal muscles and fasciae differ in women with transverse cesarean section (CS), vaginal delivery (VA) and healthy nulliparous.
The contribution of the study is mainly related to the objective estimation of fasciae and muscle “anatomy” after CS or VA.
The main (ultrasound) findings of the study are the significant alterations in both abdominal fasciae and muscle thicknesses after a CS, whereas only muscle thickness is impaired after VA.
However, the ultrasound data would be compared and analyzed before and after CS or VA.
Author Response
Thanks for your suggestion. We agree with your suggestions. It will be helpful for us to understand the abdominal status before pregnancy and after CS or VA. But it is impossible to compared and analyzed before and after CS or VA as a retrospective study. We put a limitation section and suggest that future research needs to examine the abdominal fasciae and/or muscle thicknesses before and after CS or VA.
As following:
This study had several limitations: First, being a retrospective study, many factors could not be controlled. We do not know about women’s health conditions and abdominal status before pregnancy; we are not sure that CS scar or VA is the main cause of the alterations in abdominal fasciae and/or muscle thicknesses. Future research needs to examine the abdominal fasciae and/or muscle thicknesses before and after CS or VA.
Reviewer 2 Report
This is an interesting study, with results that may have an impact on the reduction of the cesarean pandemic.The aim of the study was to assess the effect of labor - cesarean and vaginal birth- on abdominal wall thickness.
The results are relatively logical: parity in general (both vag or c delivery) affects abdominal muscles, while c sections also affect fasciae thickness, since it is incised during the operation.
I kindly have some suggestions. 1. How did you reach that sample size? 2. I suppose this was a prospective study please report it in text. 3. A limitations section would offer more value to this study. especially regarding the small sample size.
Author Response
Dear reviewer
Thanks for your suggestion. We agree with your suggestions. It is very helpful for us to improve the manuscript.
Here are the responses to your kind suggestions.
Response 1: this a retrospective study and is reported in the manuscript (line 260)
Response 2: A limitations section has been reported, considering the small sample size.
As following:
Response 3: This study had several limitations: First, being a retrospective study, many factors could not be controlled. We do not know about women’s health conditions and abdominal status before pregnancy; we are not sure that CS scar or VA is the main cause of the alterations in abdominal fasciae and/or muscle thicknesses. Future research needs to examine the abdominal fasciae and/or muscle thicknesses before and after CS or VA. Second, the current data did not include other potential mediators such as the number of childbirth, psychological and/or pathoanatomical factors. Third, the number of enrolled patients and controls was small; therefore, therefore, it is difficult to generalize these study results to all women with CS, VA. Further studies need to be replicated in larger and more diverse subject populations and alongside ultrasound fascial and/or muscular features of the lumbar, pelvic region.
Best regards!
Reviewer 3 Report
This study is of interest to those interested in the field. It presents novel results that can generate interesting recommendations for clinical practice. Even so, some aspects that can be improved to facilitate the understanding of this work are suggested
Line 34. Please include the period of time or the year since the global average of CS increased 12.4%. This may better explain the fact that they refer to
In order to improve the manuscript, I suggest that authors should explain:
- Why only one experienced operator performed ultrasound measurements of abdominal and fascial morphometry. I assume that it may be argued by the need of a highly experienced operator and the difficulty to find two operators for the recommended contrast of measurements, but it should be at least explained.
- Time between CS or vaginal delivery and measurement performance. In line 160 it is mentioned that women were recruited in postpartum. Please explain in more detail this at methods section. Recruitment and time when all measurements were performed.
- There is a difference of more than 10 years of age between nulliparous women (younger) and women who underwent a CS or had a Vaginal delivery. This should also be explained or argued. Authors say that it was difficult to recruit nulliparous women at matched age. But it can also be explained why they decided to recruit elder women with previous CS or VA. It may be understandable the difficulty to recruit the sample, but it should at least be mentioned.
Lines 125, 132, 146,148. Authors mention “cohorts”. For this type of study I suggest that it should be used the term “groups”. There is only one time point measurement and not follow up is mentioned. Otherwise please explain
Authors should explain in detail if there is a follow-up for pain assessment or if there is only one time point for the assessment, in such case explain when. Now it is confusing and probably could be better explained.
Line 254. Recommendation for women to attend exercise programs in immediate post-partum period should be recommended with caution. Pelvic floor may suffer if exercises are not those appropriate. Otherwise, it should be made an explicit mention or recommendation related to this fact.
Author Response
Dear reviewer
Thanks for your suggestion. We agree with your suggestions. It is very helpful for us to improve the manuscript.
Here are the responses to your kind suggestions.
Response 1: Line 34:
Based on data from 121 countries, the global average CS rate increased 12.4% (6.7% -19.1%) with an average annual rate of increase of 4.4% between 1990 and 2014
Response 2: Abdominal muscles and fascial thickness were measured on Esaote MyLab Seven ultrasound machine (Esaote SpA, Genova, Italy) with 37-mm linear array transductors, 6–18 MHz, by the same experienced operator in physical and rehabilitation medicine with 5 years’ experience in ultrasound skeletal-muscle imaging (Figure 1) since our previous study confirmed the good inter-rater reliability indicated that ultrasound imaging is a reliable instrument for evaluating the abdominal muscles/fasciae thicknesses [12]
the reference:
Pirri C, Todros S, Fede C, et al. Inter‐rater reliability and variability of ultrasound measurements of abdominal muscles and fasciae thickness. Clinical Anatomy. 2019; 32: 948-960.
Response 3:
We recruited the transverse CS and VA primiparous women according to the following criteria: first-born children, delivery by transverse CS or VA between 2001 and 2019, at least two years after delivery, not having taken part in any rehabilitation program and not having abdominal surgery or trauma in the post-partum period.
After giving ultrasound imaging evaluation, participants were asked to complete a short questionnaire to collect data about pain intensity, duration and location in the post-partum period. Pain intensity was measured with a visual analog scale (VAS) (mild: ⩽ 3.4, moderate: 3.5 to 7.4, severe: ⩾ 7.5). They were asked how bad the pain was in four locations: at the scar and in the areas of the abdominal wall, lower back and pelvic floor. They were also asked how long this pain had lasted after CS and VA. (acute pain: <1month, subacute pain:1-3 months, chronic pain: >3 months)
Response 4: it was difficult to recruit nulliparous women at matched age
Because the enrolled criteria: first-born children, delivery by transverse CS or VA between 2001 and 2019, at least two years after delivery, not having taken part in any rehabilitation program and not having abdominal surgery or trauma in the post-partum period. According the enrolled criteria, the recruit CS or VA women (relatively speaking) are old and it was difficult to recruit nulliparous women at matched age
Response 5: Lines 125, 132, 146,148.
The “cohorts” has been changed to the term “groups”
Response 6: there is only one time point for pain assessment in this study. We have explained in the methods section.
Primiparous participants (CS, VA) were then asked to complete a questionnaire on one time point data about pain intensity, duration and location felt to during the post-partum period.
Response 7: Line 254.
Yes, we agree that pelvic floor may suffer if exercises are not those appropriate, but, it can be useful that appropriate exercise programs in the post-partum period after the pelvic floor condition permitting
As following:
It would be useful for women to be monitored before pregnancy to assess their abdominal wall status and then to encourage them to attend appropriate exercise programs in the post-partum period after the pelvic floor condition permitting, to restore good functioning and to prevent them from new-onset complications in the following years.
Response 8: Considering the small size of this study and other limitation, we reported a limitations section, as following:
This study had several limitations: First, being a retrospective study, many factors could not be controlled. We do not know about women’s health conditions and abdominal status before pregnancy; we are not sure that CS scar or VA is the main cause of the alterations in abdominal fasciae and/or muscle thicknesses. Future research needs to examine the abdominal fasciae and/or muscle thicknesses before and after CS or VA. Second, the current data did not include other potential mediators such as the number of childbirth, psychological and/or pathoanatomical factors. Third, the number of enrolled patients and controls was small; therefore, therefore, it is difficult to generalize these study results to all women with CS, VA. Further studies need to be replicated in larger and more diverse subject populations and alongside ultrasound fascial and/or muscular features of the lumbar, pelvic region.
Best regards!